# Association between the Thickness of Lumbar Subcutaneous Fat Tissue and the Presence of Hernias in Adults with Persistent, Non-Traumatic Low Back Pain

Jorge Adolfo Poot-Franco [1,2], Anuar Mena-Balan [3], Adrian Perez-Navarrete [2], Osvaldo Huchim [4], Hugo Azcorra-Perez [5] and Nina Mendez-Dominguez [2,*]

1 Facultad de Medicina, Universidad Autónoma de Yucatán, Merida 97000, Mexico; jorgepf2013@hotmail.com
2 Hospital Regional de Alta Especialidad de la Peninsula de Yucatán, IMSS-BIENESTAR, Merida 97300, Mexico
3 Hospital Galenia, Cancun 77505, Mexico; anuarkin@icloud.com
4 Escuela de Medicina, Universidad Anahuac, Naucalpan 52786, Mexico; ohuchim@anahuac.mx
5 Centro de investigaciones Silvio Zavala, Universidad Modelo, Merida 97305, Mexico; hugoazpe@hotmail.com
* Correspondence: nina.mendez@salud.gob.mx; Tel.: +52-9999427600

**Abstract:** We aimed to analyze the association between the average lumbar subcutaneous fat tissue thickness (LSFTT) at each intervertebral level and the presence of hernias in patients with low back pain from an insurance network hospital in Mexico. This observational prospective study included 174 patients with non-traumatic lumbago who underwent magnetic resonance imaging with a 1.5T resonator. Two independent radiologists made the diagnosis, and a third specialist provided a quality vote when needed. The sample size was calculated with a 95% confidence interval using random order selection. Anonymized secondary information was used. Percentages and means with confidence intervals were tabulated. The area under the curve, specificity, and sensitivity of LSFTT were calculated. A regression analysis was performed to analyze the presence of hernias with LSFTT using each intervertebral level as a predictor. The odds of herniation at any intervertebral level increased directly with LSFTT. The average LSFTT predicted the overall presence of hernias; however, the LSFTT at each intervertebral level better predicted hernias for each intervertebral space. The area under the curve for LSFTT in predicting hernias was 68%. In conclusion, the average LSFTT was associated with the overall presence of hernias; patients with more hernias had higher LSFTT values.

**Keywords:** low back pain; hernia; subcutaneous fat; magnetic resonance imaging; intervertebral disc displacement; herniated disc

## 1. Introduction

Low back pain is a condition that influences the activities of daily living and results in absence from work and functional limitations; moreover, it is a clinical problem and a diagnostic challenge given the variety of etiologies that can cause it [1]. Low back pain is the leading cause of limitation [2]. Among the wide variety of etiologies of low back pain, herniated discs have been postulated as an important cause [3]. Some authors have suggested a link between overweight/obesity and an increased risk of low back conditions, including herniated discs, suggesting the possibility of predicting the risk of hernia using adiposity assessment [4–6].

Obesity is related to the hyperplasia and hypertrophy of fat tissue. Body mass index (BMI), an indirect indicator of obesity, has been identified as an independent risk factor for the development of musculoskeletal disorder (MSD) symptoms [4]. However, in addition to not distinguishing between fatty and lean tissues, BMI does not represent the percentage or distribution of body fat. Since lumbar spine sections focus on a specific segment, their investigation can be used to morphometrically assess local subcutaneous adipose tissue [7,8]. Özkan-Ekşi [9] validated and employed the measurement of lumbar subcutaneous fat

tissue thickness (LSFTT) using MRI parameters to assess body fat percentage, concluding that LSFTT at upper lumbar levels could predict severe intervertebral disc dysplasia (DD) better than BMI. Gürkan et al. [10] found that women and men with subcutaneous fat index cut-off values of >8.45 mm and >9.4 mm, respectively, had significantly higher rates of spinal degeneration; nevertheless, it remains unknown whether the reference value is applicable to populations with a high prevalence of overweight and obesity. Kızılgöz et al. [11] concluded that was no relationship between LSFTT and intervertebral disc herniations; however, it remains unexplored whether LSFTT at each intervertebral space or the average LSFTT of all intervertebral spaces is more reliable for the prediction of hernia presence.

Given that Mexico ranks second worldwide in the prevalence of obesity in the adult population and that more than 70% of the adult population is overweight [12,13], we aimed to establish how hernias are associated with LSFTT as an indicator of localized fat tissue. Therefore, this study aimed to analyze the association between the average LSFTT at each intervertebral level and the presence of hernias in a series of patients treated between 2020 and 2022 at an insurance network hospital in Quintana Roo, Mexico.

## 2. Materials and Methods

### 2.1. Study Design

This observational prospective study comprises imaging studies of adult patients attending consultations due to chronic, persistent low back pain. Patients were included if their pain was unrelated to trauma or pregnancy and persisted for over three months even when treated. The exclusion criteria were as follows: pregnancy at the time of consultation; low back pain caused by central nervous system infections; and low back pain of traumatic, congenital, or postsurgical origin. Even when protrusions generated symptomatology, they were excluded from the present study.

### 2.2. Patients

To establish the sample size, we employed a contrast of independent means, establishing a confidence of 95% and a maximum error of 0.05: $n = (2 \times S^2)/D^2 \times (Z_{(\alpha/2)} \times Z_{\beta})^2 = 174$, where $n$ is the sample, D is the difference, and $S^2$ is the variance.

### 2.3. Sample Selection

Using the Research Randomizer application, a list was generated that included numbers ranging between 1 and 1000, and it represented the known population size of eligible patients at an insurance network hospital between 2020 and 2022. The numbers were matched by statistics department staff who were blind to the matching of the corresponding numbers of the MRI images. The personal medical records of adult patients diagnosed with persistent, non-traumatic low back pain were anonymized and coded by technical staff at the hospital between 2020 and 2022. We received an anonymized database and encoded it in spreadsheet format, and the MRI images were linked to a unique code corresponding to the database for the later transformation of variables using Stata 14 (StataCorp, College Station, TX, USA).

### 2.4. Image Mode

Despite the possible presence of hernias along the entire spinal column, only patients with low back pain and lumbar hernias were included in this study. The images used for the calculation of LSFTT were obtained by following the standard protocol for lumbar spine imaging, which involved using a 1.5T resonator (Signa, Siemens Medical Systems Siemens, Erlangen, Germany) and an image archiving and transmission system. The patients were placed in the supine position. Sagittal and axial images were acquired using the fat saturation technique for T1- and T2-weighted MRI. The following parameters were used: a repetition time of 6029 ms; echo time (TE) of 137 ms; field of view of 280 mm; cut thickness of 4 mm; and voxel volume of 0.9 mm × 0.9 mm × 4 mm. LSFTT was measured on T1-weighted sagittal images from L1-L2 to L5-S1 in all patients. The MRIs received

included diagnostic impressions; moreover, two blinded independent radiologists with up to ten years of experience in MRI and five years of experience in musculoskeletal imaging provided reports on fat thickness measurements and the presence/absence of hernia at each lumbar spine level. The measurements were registered and averaged for each patient at each level.

Regarding the presence of hernias, a third experienced radiologist (once the mentor of radiologists 1 and 2) provided a quality vote when discrepancies arose; the resulting variables are expressed as nominal and dichotomous variables, indicating the presence/absence of hernia at each level and in the lumbar spine. The total number of hernias was also recorded as a separate discrete variable for each patient. In addition to the reasons for the consultation and the indication of MRI, every image file was labeled with the patient's age, as a numeric discrete variable, and sex, as a categorical variable.

### 2.5. Subcutaneous Fatty Tissue Thickness Measurement

Lumbar subcutaneous adipose tissue thickness was measured as the antero-posterior distance considering the tip of the supraspinous ligament as a landmark. Once the images were obtained, RadiAnt DICOM viewer 2023.1 software (Poznań, Poland) was used to delineate and segment the subcutaneous fat layer in each magnetic resonance image slice, and the edges of the fat in each image were manually traced in each intervertebral lumbar space and expressed as a numeric variable in millimeters; subcutaneous fat tissue thickness lumbar levels were summed and averaged for each patient over two continuous numerical variables.

### 2.6. Statistical Analysis

Descriptive statistics included average age and distribution by sex, the percentage of patients affected, and a graph indicating the average fat tissue in millimeters for patients with and without herniated discs in general and at each lumbar level.

Owing to the nature of the variables, a logistic regression model was used to observe the association between the measurement of LSFTT and the development of hernias at each level and any level using Stata 14 software. An interaction regression model was developed with subcutaneous fat thickness at each level and in sum to predict hernias, indicating the variance explained as Pseudo R2 and $p < 0.05$. To explore the accuracy of the considered cut-off values of the sum of subcutaneous lumbar fat for suspecting hernia, the area under the curve was determined by obtaining the sensitivity and specificity and plotting a receiver operating characteristic curve.

## 3. Results

### 3.1. Age and Subcutaneous Fatty Tissue Thickness of the Participants

This study included 174 participants (87 women and 87 men) with a mean age of $47.05 \pm 1.55$ years. The average of each level was 26.25, and the average LSFTT intervertebral level is represented and tabulated along with the descriptive statistics in Table 1.

**Table 1.** Descriptive statistics from magnetic resonance images from adult patients with chronic, persistent, non-traumatic low back pain (*n* = 174).

| Variable Description Mean | General | | Female | | Male | |
|---|---|---|---|---|---|---|
| | Mean | Std. Dev. | Mean | Std. Dev. | Mean | Std. Dev. |
| Age (years) | 47.05 | 1.55 | 48.77 | 1.50 | 45.31 | 1.58 |
| | | | | | | Thickness (mm) |
| L1-L2 LSFTT | 26 | 44 | 23 | 42 | 28 | 45 |
| L2-L3 LSFTT | 34 | 48 | 31 | 46 | 38 | 49 |
| L3-L4 LSFTT | 52 | 50 | 49 | 50 | 55 | 50 |
| L4-L5 LSFTT | 77 | 42 | 79 | 41 | 74 | 44 |
| L5-S1 LSFTT | 74 | 44 | 75 | 44 | 73 | 45 |

**Table 1.** *Cont.*

| Variable Description Mean | General | | Female | | Male | |
|---|---|---|---|---|---|---|
| | **Mean** | **Std. Dev.** | **Mean** | **Std. Dev.** | **Mean** | **Std. Dev.** |
| Prevalence | Percentage | Std. Dev. | Percentage | Std. Dev. | Percentage | Std. Dev. |
| HerniaL1-L2 | 21.66 | 1.08 | 21.52 | 1.08 | 21.81 | 1.09 |
| Hernia L2-L3 (mm) | 22.76 | 1.15 | 23.20 | 1.15 | 22.31 | 1.14 |
| Hernia L3-L4 (mm) | 26.62 | 1.32 | 27.34 | 1.35 | 25.89 | 1.29 |
| Hernia L4-L5 (mm) | 29.74 | 1.35 | 31.30 | 1.38 | 28.17 | 1.31 |
| Hernia L5-S1 (mm) | 30.46 | 1.22 | 32.27 | 1.25 | 28.64 | 1.17 |
| Two or more Hernias (%) | 36 | 48 | 47 | 50.21 | 25.88 | 44.06 |
| Overall Prevalence (%) | 65 | 48 | 69 | 47 | 61 | 49 |

The age of the participants and the number of hernias were not correlated (r = 0.06, $p$ = 0.415), but the participants with four hernias were older (50.3 years). For both sexes, a greater number of hernias were found in L5-S1, and a higher mean LSFTT was found in L5-S1; the mean LFSTT differed between levels (Figure 1).

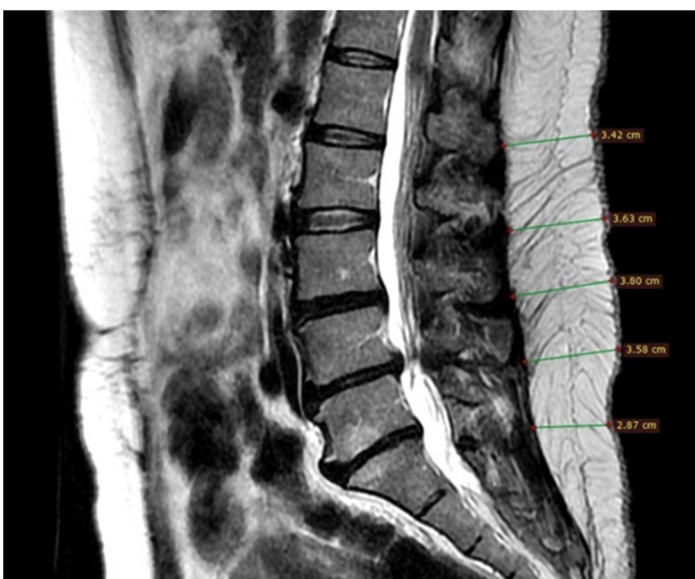

**Figure 1.** LSFTT measurements are observed at different levels of the lumbar spine, in a T2-weighted MRI in sagittal acquisition. T2-weighted AR in cross-sectional acquisition. Image obtained using Signa, Siemens medical systems.

### 3.2. Number of Hernias in Participants

The mean age of the participants was 47.05 years (SD = 15.49), without significant differences between sexes (Table 1). The overall prevalence of hernias was 65% (69% in females vs. 61% in males, $\chi^2$ $p$ < 0.001), and 36% of the patients did not have a single hernia. However, ≥2, that is, nine individuals (5.17%), had hernias at every level in the lumbar spine. Figure 2 shows the distribution along the lumbar spine in the MRI images.

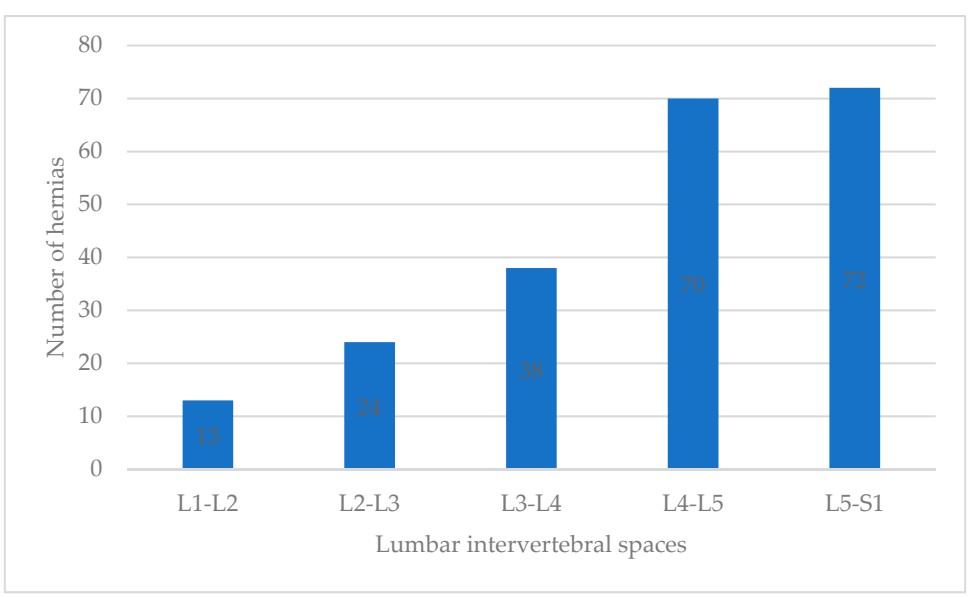

**Figure 2.** The number of hernias and mean LFSTT increased gradually from L1-L2 to L5-S1.

### 3.3. Association Analysis

The logistic regression model showed that the measurement of LSFTT in the L2-L3 lumbar segment more accurately predicted the development of hernias (odds ratio: 5.26; 95% confidence interval: 1.56–17.81), and the measurement at the L5-S1 level exhibited the strongest association with the number of hernias per patient, as shown in Table 2.

**Table 2.** Regression analysis for overall presence of hernias as dependent variable from magnetic resonance images from adult patients with chronic persistent non-traumatic low back pain (*n* = 174).

| Overall Hernia Presence | Odds Ratio | Standard Error | Z | *p* | 95% Confidence Interval | |
|---|---|---|---|---|---|---|
| Age | 0.98 | 0.01 | 1.56 | 0.118 | 0.96 | 1.00 |
| Gender | 0.65 | 0.23 | −1.21 | 0.225 | 0.33 | 1.30 |
| L1-L2 LSFTT | 4.46 | 2.80 | 2.38 | 0.017 | 1.30 | 15.24 |
| L2-L3 LSFTT | 5.26 | 3.27 | 2.67 | 0.008 | 1.56 | 17.81 |
| L3-L4 LSFTT | 4.43 | 2.77 | 2.38 | 0.017 | 1.30 | 15.07 |
| L4-L5 LSFTT | 4.62 | 2.89 | 2.45 | 0.014 | 1.35 | 15.73 |
| L5-S1 LSFTT | 4.55 | 2.84 | 2.43 | 0.015 | 1.34 | 15.43 |
| Average LSFTT L1-S1 | 0.22 | 0.13 | −2.47 | 0.140 | 0.64 | 1.73 |
| Post Hoc Hosmer–Lemeshow = 0.27 | | | | | | |
| **Number of hernias per patient** | **Coefficient** | **Standard Error** | **Z** | ***p*** | **95% Confidence Interval** | |
| L5-S1 LSFTT | 0.013 | 0.005 | 2.460 | 0.014 | 0.003 | 0.024 |

### 3.4. Specificity and Sensitivity of LSFTT with Hernia Presence

A cut-off point of ≥81 mm of the LSFTT measurement was found on the reference diagonal line, giving an area under the curve of 68.7% for the presence of hernias from the sum of LSFTT, obtaining a sensitivity of 85.45% and a specificity of 32.36% (Figure 3). Additionally, for the mean LSFTT in L1-S1, with a cut-off of ≥17.24, 66.8% of cases were predicted correctly, with a sensitivity of 80.00% and a specificity of 41.94%. Finally, using the LSFTT in L5-S1, the sensitivity was 85.45%, the specificity was 50.00%, and the percentage of correctly predicted cases was 67.1%.

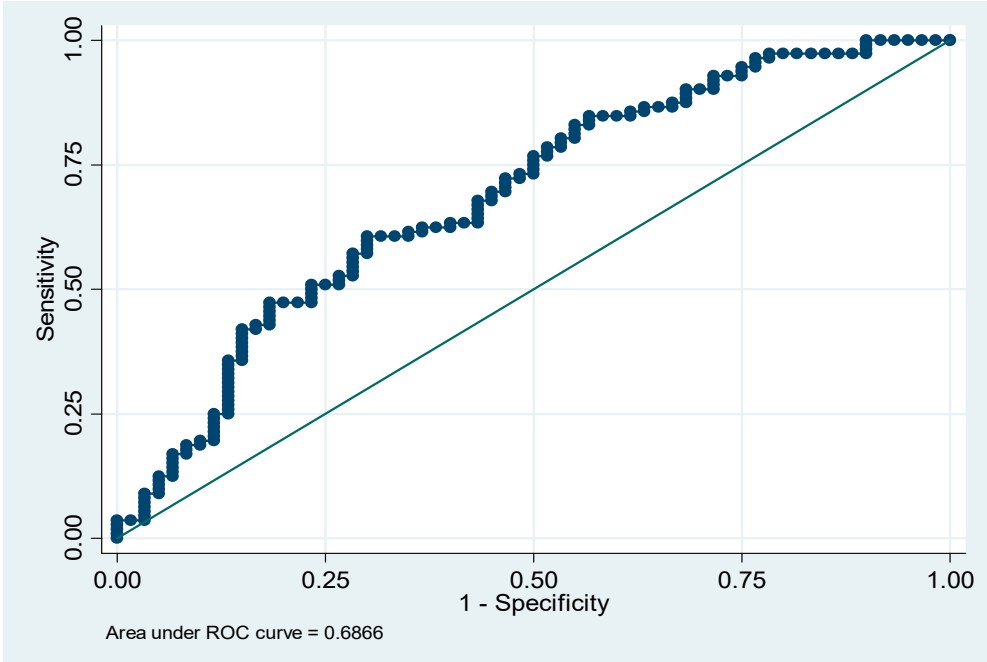

**Figure 3.** Area under the curve for presence of hernias from sum of lumbar subcutaneous fat tissue thickness. Obtained by establishing a cut-off value of ≥81 mm, sensitivity = 85.45% and specificity = 32.36.

## 4. Discussion

In this study, we found that LSFTT varied between men and women; however, the distribution was similar. In the studied patients, lumbar pain was related to hernias in 69% of women and 61% of men. The presence and number of hernias correlated with the average LSFTT and LSFTT at each corresponding level in our sample.

We also found that the presence of hernias could be inferred from LSFTT in 68% of the patients with non-traumatic, chronic, and persistent low back pain. Previously, Kızılgöz et al., who assessed the L4-L5 level in a series of 102 patients, found that LSFTT and lumbar indentation values were independently distributed in patients with and without herniation but that paraspinal muscle atrophy predicted hernia [14]. We believe that the differences found may be related to the selection of a single intervertebral space. Because our study was exploratory in nature, no controls were assigned to each case.

Up to 39% of patients with low back pain will present with a herniated disc as a possible cause [15]. Some authors have reported a strong correlation between low back pain and intervertebral disc degeneration [16]. Obesity is considered a risk factor for degenerative changes in the spine, including herniated discs [17]. In a study by Güleç and Karagöz Güzey, a strong correlation was found between BMI and the probability of surgical reintervention [8], meaning that fat tissue may indicate not only the occurrence but also the complexity of hernias. Considering the high prevalence of overweight and obese individuals in the geographical region where the study was conducted [18], the results of our study show a high prevalence of 65% of patients with the presence of herniated discs, unlike the results obtained by Kizilgöz, where no relation was found between LSFTT and the presence of herniated discs [16].

Obesity measured using BMI or LFSTT is not the only risk factor for the development of lumbar hernias; however, different lumbar hernia prevalence rates have been reported depending on the area of the world where the study was conducted. Therefore, patients should be staged according to the states or countries where they reside or have lived most of their lives, as well as by their different ethnicities, races, and other socioeconomic and cultural conditions that may be risk factors for the increased presence of hernial defects in discs [19–21].

Özcan-Ekşi et al. (2018) mentioned a high percentage of changes in the intervertebral discs in men [9], unlike in our study, which had almost the same proportion of male and female patients. In 2015, Teichtahl et al. [22] found that high multifidus fat [>50%] correlated with proneness to high-intensity pain/disability after adjusting for age, sex, and BMI, with a significant odds ratio of 12.76. In addition, in the mentioned study [9], the sample of 178 patients had a mean age of 35.8 years, and the subcutaneous fat in millimeters was 14, 14.2, 18.7, 25.6, and 30.2 in L1-L2, L2-L3, L3-L4, L4-L5, and L5-S1, respectively. These results are different from those obtained in our study, which included a similar sample size; however, the mean patient age was 47.13 years, and the mean LSFTT in millimeters was 21.6, 22.7, 26.6, 29.7, and 30.4 in L1-L2, L2-L3, L3-L4, L4-L5, and L5-S1, respectively, showing a larger LSFTT deposit in our sample, except for in L5-S1, which is similar in both studies.

According to Berikol et al. in 2022 [10], the cut-off value for increased LSFTT that determines the presence of disc degeneration is 9.4 mm and 8.45 mm in men and women, respectively. However, in our study, 98.8% of the patients presented values exceeding these cut-off points, which may be, at least in part, related to the differences in overweight and obesity prevalence, and we propose that no cut-off point may be considered generalizable given the phenotypic variability across different populations.

Adipose tissue in the lumbar area is either a cause or consequence of lumbar hernias, and this may need to be further studied in cohort designs. Nevertheless, the accumulation of fat tissue in the central region of the body, as a result of a chronic energy imbalance, and the consequent increase in body weight may increase the risk of lumbar hernias due to the pressure exerted on the intervertebral spaces; however, it is reasonable that hernias occur due to causes other than excess weight and their secondary effects. In such cases, conditions may coexist and produce pain and reduce mobility [23], and after some time, they may affect the patient's capacity for physical activity, promote fat accumulation, and facilitate central obesity. In our study, we limited our findings to show the variation in the distribution and amount of lumbar adipose tissue in adults in association with the presence, number, and location of hernias in the lumbar spine.

The present study has some limitations. First, there was a lack of contributing variables, which prevented the development of a more comprehensive model to predict hernias. Second, all images were obtained in patients in the supine position due to the resonator model, which is the ideal resonator for upright MRI of the lumbar spine [22]; however, it may not be ideal for additional analyses or for assessing low back pain in patients who are unable to sit. In conclusion, from our study, it can be concluded that the average LSFTT was associated with the overall presence of hernias; patients with more hernias also had a greater LSFTT.

## 5. Conclusions

Lumbar pain was related to hernia in 69% of women and 61% of men in a sample obtained from an insurance network hospital in Quintana, Roo, Mexico. The presence and number of hernias correlated with the average lumbar subcutaneous fat tissue thickness at each corresponding level, and this measure may have a good sensitivity but a low specificity for the studied population.

**Author Contributions:** N.M.-D. and A.M.-B.; methodology, N.M.-D., A.P.-N. and A.M.-B.; software, J.A.P.-F. and N.M.-D.; validation, N.M.-D., A.M.-B. and H.A.-P.; formal analysis, N.M.-D., A.M.-B., A.P.-N. and J.A.P.-F.; investigation, J.A.P.-F., N.M.-D. and H.A.-P.; resources, J.A.P.-F., N.M.-D. and H.A.-P.; data curation, N.M.-D., O.H., A.P.-N. and J.A.P.-F.; writing—original draft preparation, J.A.P.-F., N.M.-D., O.H. and A.P.-N.; writing—review and editing, N.M.-D., A.M.-B., O.H., H.A.-P., A.P.-N. and J.A.P.-F.; visualization, N.M.-D. and H.A.-P.; supervision, N.M.-D. and A.P.-N.; project administration, N.M.-D. All authors have read and agreed to the published version of the manuscript.

**Funding:** This research received no external funding.

**Institutional Review Board Statement:** The study was conducted in accordance with the Declaration of Helsinki and approved by the Institutional Review Board Hospital (protocol code AH23001 and 2022/04/01).

**Informed Consent Statement:** All procedures performed in studies involving human participants were in accordance with the ethical standards of the institutional and/or national research committee and with the 1964 Helsinki declaration and its later amendments or comparable ethical standards. Approval from the Institutional Review Board No. AH23001 was obtained and in keeping with the policies for a retrospective review, informed consent was not required.

**Data Availability Statement:** Available at ResearchGate profile pages of authors and upon reasonable request via e-mail.

**Conflicts of Interest:** The authors declare no conflicts of interest.

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
