# Peer review of "Association between the Thickness of Lumbar Subcutaneous Fat Tissue and the Presence of Hernias in Adults with Persistent, Non-Traumatic Low Back Pain"

_tomography, doi:10.3390/tomography10020022_

Round 1

Reviewer 1 Report (Previous Reviewer 1)

Comments and Suggestions for Authors

Introduction

It needs restructuring; sentences are thrown all over the place, and there is no consistency or flow. It needs to be rewritten. Includes information in the wrong place.

L59: "between 2020 and 2022." out of place

Materials and Methods 

L61-62 "2.1. Study design. In this observational prospective study comprises imaging studies of 61 adult patients consulting due to chronic, persistent lower back pain." It makes no sense.

L70: What is S=? What is D=?

L83: What is the standard protocol?

L95: Please define "experienced radiologist" and report the others' experience level.

Discussion

It also needs to be rewritten.

L179: repetition of the information from the methods.

L187-197: BMI, W and H from patients, level of activity, type of work...data is missing from this study!? Please include the missing data.

L241-242: This is the opposite of the affirmation in L223-224.

Comments on the Quality of English Language

Poorly written paper. The introduction and discussion need to be rewritten. There is a difference in using an Editing service and the context included in the manuscript as well as the structure.

Author Response

Dear reviewer, we thank the time and care you invested in suggesting improvements to our manuscript, we hope we have addressed accordingly, please find our point-by-point response. We marked changes in yellow in the manuscript.

Introduction. We have restructured and undergone two-step professional. editing, first by our institutional copy editor and also by MDPI services.

L59: We have corrected, thank you

L61-62 "2.1. We have corrected, thank you

L70: We have clarified, thank you

L83: Our wording was incorrect, We have corrected, thank you.

L95: we meant the professor of the other two radiologists, we have rephrased

We have reformulated and rewritten the discussion.

L179: we deleted the repeated information.

L187-197: we have clarified the point why we were mentioning such parameters, we did not have them.

L241-242 and congruence with L223-224. We have rephrased and corrected.

Reviewer 2 Report (Previous Reviewer 2)

Comments and Suggestions for Authors

I consider that the authors have done a great job of researching and reworking the original manuscript; I congratulate you. And I think they have responded correctly to the reviewers' instructions. Thanks for the effort and availability.

Kind regards

​

Author Response

Thank you, dear reviewer. He sent the manuscript in the newest version for professional English Editing.

Reviewer 3 Report (New Reviewer)

Comments and Suggestions for Authors

Dear authors,

Thank you for submitting your manuscript titled “Association between the thickness of lumbar subcutaneous fat tissue and the presence of hernias in adults with persistent, non-traumatic low back pain” for review.

I have identified several areas where the manuscript could benefit from further enhancements. Below are my detailed suggestions:

-          Lines 41-42: The spine MRI cannot “estimate and quantify the location of adipose tissue deposition”. Since the lumbar spine MRI sections are focused on a specific segment, this investigation can morphometrically assess the local subcutaneous adipose tissue. Please reconsider.

-          Line 91: 5-10 years

-          Lines 92-93 and 96-97 - similar statement

-          Lines 102-104: I noticed a discordance between the described measurement landmark (the tip of the spinous process) and the landmark from which the measuring axis begins (Figure 1). Please consider the supraspinous ligament as a landmark.

-          Line 107: “the amount of fat present in each intervertebral lumbar space” – measured fat is not in intervertebral space. Please reformulate.

-          Line 131: the Figure 1 does not support the affirmation.

-          Table 1: the dimensions of the hernias do not correspond to reality (mean > 20 mm?). In my opinion, the table must be modified, possibly divided into two or three sections.

-          Line 186: the reference [15] must be placed before one's own opinions

-          Line 196: reference is missing

-          Lines 209-210: reformulation is necessary; the mentioned study is Teichtahl et al.[22] not [9]

-          Lines 194, 206 and 223: consider to change “this study” with “our study” –

-          Lines 43, 47, 50, 182, 205, 207, 217: consider placing the reference immediately after the primary author of the study. Example: Kızılgöz et al. [11]

-          For an easier reading and understanding of the subject, I recommend using only one term, especially as an abbreviation: LSFTT or SFTT. Including in Table 1, both terms appear with the same meaning.

I hope these suggestions will be helpful in strengthening your manuscript and better conveying the important research you have undertaken. The final goal is to improve the overall clarity of the message to help the reader understand this interesting topic. Overall, my peer review is a major revision.

Looking forward to seeing the revised version of your work.

Best regards.

Author Response

I have identified several areas where the manuscript could benefit from further enhancements. Below are my detailed suggestions:

Thank you, dear reviewer, we are grateful, you have truly helped us finding aspects needing improvement in the manuscript.

We have sent our manuscript for editing by an institutional colleague and with MDPI services, after correcting aspects you kindly pointed, you can find the changes highlighted in yellow in te newest version.

Please find pont-by-point responses to every aspect you mentioned.

  •          Lines 41-42: The spine MRI cannot “estimate and quantify the location of adipose tissue deposition”. Since the lumbar spine MRI sections are focused on a specific segment, this investigation can morphometrically assess the local subcutaneous adipose tissue. Please reconsider.
  • Thank you we followed your kind suggestion.
  •          Line 91: 5-10 years
  • Thank you, we have corrected.
  •          Lines 92-93 and 96-97 - similar statement
  • Thank you, we have corrected
  •          Lines 102-104: I noticed a discordance between the described measurement landmark (the tip of the spinous process) and the landmark from which the measuring axis begins (Figure 1). Please consider the supraspinous ligament as a landmark.
  • Thank you, we followed your kind recommendation 
  •          Line 107: “the amount of fat present in each intervertebral lumbar space” – measured fat is not in intervertebral space. Please reformulate.
  • Thank you, we have corrected
  •          Line 131: the Figure 1 does not support the affirmation.
  • You are right, that specific patient did not follow such pattern, we have corrected.
  •          Table 1: the dimensions of the hernias do not correspond to reality (mean > 20 mm?). In my opinion, the table must be modified, possibly divided into two or three sections.
  • Wow! you have been really helpful, we did not notice that it was not the mean but the prevalence, we changed the layout but forgot to change the header. we have now corrected.
  •          Line 186: the reference [15] must be placed before one's own opinions.
  • Thank you, dear reviewer, we have corrected.
  •          Line 196: reference is missing
  •  
  • Thank you, dear reviewer, we have corrected.
  •  
  •          Lines 209-210: reformulation is necessary; the mentioned study is Teichtahl et al.[22] not [9]
  •  
  • Thank you, dear reviewer, we have corrected.
  •  
  •          Lines 194, 206 and 223: consider to change “this study” with “our study” –
  •  
  • Thank you, dear reviewer, we have corrected.
  •  
  •          Lines 43, 47, 50, 182, 205, 207, 217: consider placing the reference immediately after the primary author of the study. Example: Kızılgöz et al. [11]
  •  
  • Thank you, dear reviewer, we have corrected.
  •  
  •          For an easier reading and understanding of the subject, I recommend using only one term, especially as an abbreviation: LSFTT or SFTT. Including in Table 1, both terms appear with the same meaning.
  •  
  • Thank you, dear reviewer, we have corrected.
  •  

Round 2

Reviewer 1 Report (Previous Reviewer 1)

Comments and Suggestions for Authors

A few minor corrections.

Introduction

L35-37 and L45-48: A paragraph cannot be just one sentence. Please correct this.

Discussion

L231: "In our study, we limited our findings to showing the variation in..." Please correct "to show" not "to showing".

Comments on the Quality of English Language

Improved but needs improvement in the introduction. One sentence cannot be a paragraph.

Author Response

Thank you, dear reviewer, we made the changes you kindly suggested. Now the manuscript is clearer and more readable.

Please find the changes in the newest version highlited in yellow.

Reviewer 3 Report (New Reviewer)

Comments and Suggestions for Authors

Dear authors,

Thank you for re-submitting your manuscript titled “Association between the thickness of lumbar subcutaneous fat tissue and the presence of hernias in adults with persistent, non-traumatic low back pain”.

After the changes made, I consider that the manuscript meets the conditions for publication.

Author Response

Thank you, dear reviewer, for your insightful comments.

This manuscript is a resubmission of an earlier submission. The following is a list of the peer review reports and author responses from that submission.

Round 1

Reviewer 1 Report

Comments and Suggestions for Authors

Introduction:

Looks like sentences thrown all over the place: no consistency or flow. It needs to be rewritten. Includes information in the wrong place, i.e. L58-60.

Material and Methods: 

L68-70: Did you account for dro-outs?

L74: What do you mean by "using the MRI results"?

L112-119: What software did you use? What p-value?

Where is Figure 1? Figure 1 is a picture in text.

Discussion, same as for the introduction.

L120-125: Should not be placed there.

Comments on the Quality of English Language

Very poor written paper. I suggest rejection and resubmission after substantial revision.  The authors did not even take the time to add author affiliations properly or add author contributions.

Reviewer 2 Report

Comments and Suggestions for Authors

I congratulate you on your research, which I consider interesting and well carried out.

I would like to ask you some questions and recommendations:

1) In the Keywords, I advise adding MesH terms, such as: "Intervertebral Disc Displacement", or "Herniated Disc". This way your article will have more visibility in the databases.

2) In the phrase "Gürkan et al. established the SFI cut-off values...", it would be convenient to also put the full name that SFI means, when it is used for the first time in the text.

3) Because the symptoms of low back pain may or may not be caused by hernias in each patient studied, where it says:

2.4. Image mode. Despite the possible presence of hernias along the entire spinal column, 81 only symptomatic lumbar hernias were included in this study.

I think it's better to say:

2.4. Image mode. Despite the possible presence of hernias along the entire spinal column, 81 only patients with low back pain and lumbar hernias were included in this study.

4) When you refer to herniated discs, could you specify if you differentiated between herniated discs and protrusions?

5) On line 132 it states: "For both sexes, a 132 greater number of hernias were found in L4-L5..."

But in Fig.2 and Table 1 L5-S1 appears as the level with the most hernias. Can you clarify this point?

​6) On line 192 it states: "Up to 39% of patients with lower back pain will present with a herniated disc as a cause." I consider it more correct to state: "Up to 39% of patients with lower back pain will present with a herniated disc as a possible cause"

​

Reviewer 3 Report

Comments and Suggestions for Authors

Please see the file

Comments on the Quality of English Language

Ok, minor